# Predictors of Death in Patients with Neonatal Sepsis in a Peruvian Hospital

**DOI:** 10.3390/tropicalmed7110342

**Published:** 2022-10-31

**Authors:** Dariela Vizcarra-Jiménez, Cesar Copaja-Corzo, Miguel Hueda-Zavaleta, Edgar G. Parihuana-Travezaño, Maykel Gutierrez-Flores, Marco Rivarola-Hidalgo, Vicente A. Benites-Zapata

**Affiliations:** 1Facultad de Ciencias de la Salud, Universidad Privada de Tacna, Tacna 23003, Peru; 2Red Asistencial Ucayali EsSalud, Ucayali 25003, Peru; 3Hospital III Daniel Alcides Carrion EsSalud, Tacna 23000, Peru; 4Hospital Hipólito Unanue de Tacna, Tacna 23003, Peru; 5Unidad de Investigación para la Generación y Síntesis de Evidencias en Salud, Universidad San Ignacio de Loyola, Lima 15024, Peru

**Keywords:** neonatal sepsis, newborn, mortality, premature, Peru, risk factors, sepsis, thrombocytopenia

## Abstract

Reducing neonatal mortality is a global challenge. This study’s objective was to determine the predictors of mortality in patients with neonatal sepsis. The study was a retrospective cohort study in a Peruvian hospital from January 2014 to April 2022. Neonates diagnosed with sepsis were included. To find predictors of mortality, we used Cox proportional regression models. We evaluated 288 neonates with sepsis; the median birth weight and hospitalization time were 3270 g and seven days, respectively. During follow-up, 18.4% did not survive, and the most common complications were jaundice (35.42%), respiratory distress syndrome (29.51%), and septic shock (12.5%). The most isolated bacteria were *Klebsiella pneumoniae*. The risk factors associated with higher mortality were prematurity (aHR = 13.92; 95% CI: 1.71–113.51), platelets <150,000 (aHR = 3.64; 1.22–10.88), creatinine greater than 1.10 (aHR = 3.03; 1.09–8.45), septic shock (aHR = 4.41; 2.23–8.74), and admission to IMV (aHR = 5.61; 1.86–16.88), On the other hand, breastfeeding was associated with a lower risk of death (aHR = 0.25; 0.13–0.48). In conclusion, we report a high incidence of death and identify clinical (prematurity, septic shock, admission to IMV) and laboratory characteristics (elevated creatinine and thrombocytopenia) associated with higher mortality in patients with neonatal sepsis. Breastfeeding was a factor associated with survival in these patients.

## 1. Introduction

Combating infant and neonatal mortality is a global challenge [1]. Although mortality in children under five years of age has decreased from 5 million in 1999 to 2.5 million in 2017 [2], a high mortality rate in newborns persists [3], especially in developing countries such as those in Latin America [4]. One of the leading causes of neonatal mortality is sepsis, which has been described as a clinical syndrome characterized by signs and symptoms of infection with or without accompanying bacteremia [5]. Mortality due to neonatal sepsis varies between 9% and 65% [6], depending on factors such as gestational age, maternal characteristics, and the associated bacteria.

In developed countries, despite the advanced strategy of preventing and managing neonatal sepsis, a mortality rate between 5% and 20% occurs and causes significant disability in patients, despite early initiation of treatment [7,8,9,10]. In contrast, countries in the Middle East and Africa report mortality rates ranging between 10% and 30%. In emerging countries such as Egypt and South Sudan, neonatal mortality rates of over 50% have been reported [11,12]. Previous studies in Peru report a high neonatal sepsis mortality rate (21.6%), similar to countries in the Middle East and Africa. Despite the advances and socioeconomic development of Peru, neonatal sepsis is the second cause of neonatal mortality in this country, with neonatal mortality being the main component of general mortality in all Peruvian children under one year of age (66.6%) and children under five years of age (55.6%) [13,14]. On the other hand, the complications of sepsis are associated with more extended hospital stays, care costs, poor neurological development, and inadequate growth in early childhood [15,16,17], recognizing this disease as a serious public health problem [5].

Although inflammatory markers with adequate sensitivity and specificity have been evaluated to identify infected newborns early [18,19], in Peru—and more widely in Latin American countries—information on predictors of death in newborns with sepsis is lacking. Especially in the context of scarce resources, these limit the analysis of more specific inflammatory markers because they are more expensive [20]. In this sense, this research aimed to determine predictors of death in patients with neonatal sepsis treated at a reference hospital in southern Peru.

## 2. Materials and Methods

### 2.1. Environment and Research Design

A retrospective cohort study was designed with data collected from the medical records of the Hipólito Unanue Hospital in the Tacna (HUHT) region, Peru. This reference hospital in southern Peru has a neonatal intensive care unit (NCIU) with twenty-five beds, ten fixed incubators, and two transport incubators, in addition to six neonatal mechanical ventilators (VMI). The study period was from 1 January 2014 to 30 April 2022. The manuscript was written following the guidelines of Strengthening the Reporting of Observational Studies in Epidemiology (STROBE) [21].

### 2.2. Population and Sample

The study population included the medical records of neonates who were admitted to the neonatal intensive care unit with a diagnosis of neonatal sepsis, which was defined as the presence of more than one of the following characteristics: fever (temperature >38 °C) or hypothermia (temperature <36 °C), rapid breathing (>60 breaths per minute), severe chest indrawing, poor feeding, seizure, lethargy, or unconsciousness; along with two of the hematologic criteria: total leukocyte count <5000 or >12,000 cells/m^3^, absolute neutrophil count <1500 cells/mm^3^ or >7500 cells/mm^3^, erythrocyte sedimentation rate (ESR) >15/1 h, and platelet count <150,000 or >440,000 cell/mm^3^ for late sepsis [22], and for early sepsis the Rodwell criteria [23]; and that the neonates also had a definitive result during follow-up (hospital discharge or death). Patients still hospitalized when the study was carried out were excluded, in addition to medical records that did not have the necessary information to know the variables of interest.

To calculate the sample size, we used the study by Bekele T. et al., [22]; the independent variable (exposure) was gestational age <37 weeks, and the outcome was death. In their study, 60% of patients with gestational age <37 weeks died during follow-up, compared to 40% mortality in those with age ≥37 weeks. Likewise, a non-exposed/exposed ratio of 2.98 (164/55) was reported with these parameters: a confidence level of 95%, and a power of 80%. We calculated a sample size of 288, to which 10% was added for possible medical records with insufficient data, finally calculating a total of 310 records.

### 2.3. Data Collection and Variable Definition

The general register of the HUHT neonatology service was used. We evaluated all patients hospitalized between 1 January 2014 and 30 April 2022. Once identified, we collected information from the patient’s physical medical records from their hospitalization in the NICU, such as clinical characteristics, laboratory, and treatment. Clinical and laboratory features were collected from the time when sepsis was suspected. The attributes in the treatment were taken during the hospital stay, and the patients were followed up until the clinical outcome (discharge or death). Two investigators collected the data, double data entry was performed (by different investigators), and a third investigator was in charge of quality control. The information was contrasted with the physical clinical history when any variation was found.

#### 2.3.1. Outcome Variables

Hospital mortality: Mortality in patients diagnosed with neonatal sepsis was evaluated as an outcome variable. This was collected according to the result recorded in the medical record until the end of the follow-up (30 April 2022).

#### 2.3.2. Exposure Variables

Clinical features: Clinical characteristics were: early sepsis (<72 h of age at diagnosis), late sepsis (>72 h of age at diagnosis), gender (male/female), gestational age (grouped into <37 weeks and >37 weeks), birth weight (grams), APGAR at 1, 5, and 10 min (<4/4–6/>7 points), temperature (<36.1/>36.1 °C), respiratory rate (<60/>60 breaths per minute), heart rate (<80/>160 beats per minute), oxygen saturation (<95/>95%), weight variance (grams), type of delivery (vaginal/cesarean section), the reason for cesarean section (fetal pathology/maternal pathology), hospital care, use of mechanical ventilation, central venous catheter, umbilical venous catheter, umbilical arterial catheter, lumbar puncture, associated congenital disease (fetal malformation/genetic disease/fetal infections/neonatal complication), breastfeeding, infant formula, and total parenteral nutrition (TPN), as well as the entire stay in the hospital and the NICU (days) [24,25,26,27].

Additional reference tests: The following laboratory tests were considered: blood culture, isolated bacteria, glucose (<150/>150 mg/dL), leukocytes (<12,000/>12,000 cells/mm^3^), platelets (>150,000/<150,000 cells/mm^3^), total neutrophils (<10,000/>10,000 cells/mm^3^), total rods (<100/>100 cells/mm^3^), immature/total index (>0.12/<0.12), hemoglobin (<13.5/>13 0.5 g/dL), hematocrit (<42/>42%), erythrocyte amplitude velocity (<15/>15%), albumin (<3.80/>3.80 g/dL), lactic dehydrogenase (< 100/>100 U/mL), C-reactive protein (<10/>10 mg/dL), creatinine (<1.10/1.10 mg/dL), CPK-MB (<10/>10 U/L), aspartate aminotransferase (AST)(<38/>38 U/L), alanine amino transferase (ALT)(<41/>41 U/L), alkaline phosphate (<300/>300 U/L), lactate (<2/>2 mmol/L), sodium (<145/>145 mmol/L), potassium (<5.50/>5.50 mmol/L), chlorine (<110/>110 mmol/L), bicarbonate (<26/>26 mmol/L), pH (<7.35/>7.35), CO2 pressure (<45/>45 mmHg), O2 pressure (<50/>50 mmHg), FiO2 supply (%), and the correlation between the oxygen pressure on the inspired fraction of oxygen (PaO2/FiO2; <400/>400), and these were evaluated upon NCIU at the hospital admission [28,29,30,31,32,33,34,35,36,37,38].

Complications during NCIU stay: During the stay in the NCIU, the complications were: jaundice, septic shock, acute respiratory distress syndrome (ARDS), acute kidney failure (AKI), multiple organ failure, and seizures; these diagnoses were established according to international criteria [39,40,41,42].

Treatment received: Treatments administered to patients at diagnosis of neonatal sepsis included: antibiotics (oxacillin, ampicillin, aminoglycosides, cefotaxime, ceftazidime, imipenem/meropenem, vancomycin, metronidazole, clindamycin, fluconazole, amphotericin B), vasopressors, corticosteroids, diuretics, bovine pulmonary surfactant, phototherapy, blood transfusion, anticonvulsants, and antipyretics.

### 2.4. Statistic Analysis

We downloaded the information in a Microsoft Excel document and then exported it for analysis in the statistical program STATA v17. The categorical variables were described in absolute frequencies and percentages; we used measures of central tendency and dispersion for the numerical variables. For the bivariate analysis between the outcome variable (mortality) and the categorical variables, we used the χ^2^ statistical test or Fisher’s exact test, as appropriate. We used the Mann–Whitney U statistical test for the numerical variables since all the variables presented an asymmetric distribution.

To determine predictors of death in patients with neonatal sepsis, we used Cox proportional hazard models. Crude (cHR) and adjusted (aHR) hazard ratios and their respective 95% confidence intervals (95% CI) were determined. Because we had a small number of events (non-survivors = 53), we ran two Cox proportional regression models to reduce overfitting between variables. The first model used the variables of gestational age, platelets, leukocytes, total immaturity index (TII), glucose, creatinine, pH, O2 pressure, and PaO2/FiO2, and the second model used the variables of birth weight, APGAR at minute, APGAR at 5 min, use of a central venous catheter, breastfeeding, invasive mechanical ventilation (IMV), presence of shock, and acute respiratory distress syndrome (ARDS). We decided to separate the variables of gestational age and birth weight since they presented collinearity between them. Finally, we decided to perform a secondary analysis of the predictors of death only in patients with early neonatal sepsis, using the same methodology as in sepsis in general. The proportionality assumptions were tested using Schoenfeld residuals. 

### 2.5. Ethics

This study was conducted following the international research ethics guidelines of the Declaration of Helsinki. The research protocol was evaluated and approved by the ethics committee of the Faculty of Health Sciences of the Private University of Tacna (identification code: 098-FACSA-UI). Informed consent was not requested due to the observational and retrospective nature of the study.

## 3. Results

A total of 310 neonates were enrolled, of which 22 were excluded (due to lack of data in the medical records). A total of 288 neonates with sepsis were analyzed; the flow diagram of the study population is shown in Figure 1**.**

### 3.1. Population Characteristics

Most of the final sample evaluated was male (50.35%), the median gestational age was 39 weeks, and the birth weight was 3270 g. The median follow-up time of the patients was seven days (IQR: 6–14)**.** Of the newborns admitted to the NCIU for sepsis, their average hospital stay was seven days; 35.42% presented jaundice, 29.51% presented respiratory distress, 223 (77.43%) neonates showed early sepsis, and 35 (12.5%) showed septic shock. Regarding the type of feeding, 57.64% received breastfeeding (Table 1).

Mortality during hospitalization was 18.40%, with a mortality rate of 1.5 deaths per 100 person/day of risk. During hospitalization, the most common complications in non-survivors were respiratory distress syndrome, acute renal failure, and septic shock. Likewise, the most common antibiotic treatment was ampicillin and aminoglycosides. On the other hand, the median APGAR score at 1 and 5 min after birth was 8 (IQR: 6–9) and 9 (IQR: 7–9) points, respectively. Finally, 57.64% of newborns received exclusive breastfeeding.

### 3.2. Bivariate Analysis according to Mortality in the Study Population

During the hospital stay, statistically significant differences were observed in mortality in laboratory results and management of patients with neonatal sepsis. It was found that hemoglobin, red blood cells, and platelet levels were lower in the group of patients who died. It was also observed that glycemia, ALT, and creatinine levels were higher in the patients who died. Of the blood cultures taken, only 9.03% were positive, and *Klebsiella pneumoniae* ESBL was the most isolated bacteria. Blood transfusion and administration of exogenous pulmonary surfactant were higher than in the group that did not survive. (Table 2).

### 3.3. Predictors of Mortality in Patients with Neonatal Sepsis

In the crude Cox regression analysis, gestational age was associated with higher mortality risk (cHR = 19.95; 95% CI: 7.85–50.69). The multivariate analysis included laboratory tests such as leukocyte levels, platelets, IT ratio, glucose, creatinine, pH, PCO2, PaFiO2, and gestational age. Gestational age was independently associated with hospital mortality (aHR = 13.92; 95% CI: 1.71–113.51). Likewise, low platelet values <150,000 (aHR3.64; 95% CI: 1.22–10.88), and high creatinine levels >1.10 (aHR = 3.03; 95% CI: 1.09–8.45) were associated with higher hospital mortality (Table 3). On the other hand, in the secondary analysis that we performed only on patients with early sepsis, these predictors of death remained constant, except for creatinine (Appendix A).

Multivariate analysis also included clinical variables such as gestational age, birth weight, APGAR at 1 and 5 min, septic shock, ARDS, use of IMV, CVC, and exclusive breastfeeding. The birth weight (aHR: 2.94; 95% CI: 1.04–8.30), use of an invasive mechanical ventilator (aHR = 5.61; 95% CI: 1.86–16.88) and septic shock (aHR = 4.41; 95% CI: 2.23–8.74) were associated with higher hospital mortality in the adjusted analysis. On the other hand, exclusive breastfeeding was associated with a higher survival probability than patients who did not use it (aHR = 0.25; 95% CI: 0.13–0.48) (Table 4). On the other hand, in the secondary analysis that we performed only on patients with early sepsis, these predictors of death remained constant (Appendix A).

### 3.4. Survival Estimated by Kaplan–Meier Curve

Patients with neonatal sepsis who did not present gestational age <37 weeks, developing septic shock, use of IMV, thrombocytopenia, and elevated creatinine level at diagnosis, presented a better survival curve. The use of exclusive breastfeeding in patients with neonatal sepsis was associated with more remarkable survival, the difference being statistically significant (Figure 2).

## 4. Discussion

In this retrospective cohort study, we found a high mortality rate from neonatal sepsis and a low rate of positive blood cultures. The most common bacteria were *Klebsiella pneumoniae* ESBL. Predictors of mortality in newborns with sepsis were gestational age of fewer than 37 weeks, thrombocytopenia, and elevated creatinine levels. Of the total (77) premature infants, 62.34% died, 20.78% were late-premature, 14.29% were moderately preterm, 31.17% were very preterm, and 33.77% were extremely preterm at birth, respectively, with the group of extremely preterm newborns presenting the highest mortality rate. The mortality rate due to neonatal sepsis was 18.40% in our study. This is comparable to other studies such as those conducted in hospitals in Ethiopia that report a mortality rate of between 18.6% to 25.5% [43,44,45], Cameroon (15.7%) [46], and southeastern Nigeria (19.4%) [47]. This similarity in neonatal mortality could be due to various factors, such as limited health resources and higher birth rates. In contrast, our mortality rate was higher than that reported in the United States (<11%) [48,49], Qatar (<6.5%) [50], and Portugal (<5.7%) [51]. This is probably because in developed countries or cities, non-preventable conditions, such as congenital anomalies, are the leading risk factor for death from neonatal sepsis. In contrast, in developing countries, newborns die from preventable conditions such as prematurity and septic shock, this being of significant impact on neonatal death [52].

The gold standard for diagnosing neonatal sepsis is isolating a bacterium through a blood culture. Taking the blood sample can be performed before or after the administration of the antibiotic and influences the results; however, positivity is still found in this method in more than 20% of patients who received antibiotics after sampling [53]. In this study, we found that only 9% of the patients presented positivity in blood cultures, and the most isolated bacteria were Klebsiella Pneumoniae ESBL. Blood cultures’ low percentage of positivity could be due to different reasons. First, our hospital does not have an automated culture system; on the other hand, the microbiology area only has 8 h shifts, which could prevent adequate processing and evaluation of the collected samples [54]. This is an important limitation in contexts of scarce economic resources in health; providing a larger budget and better management, as well as generating agreements with other institutions, could provide a possible solution to this problem common in developing countries.

### 4.1. Neonatal Mortality Predictors

Prematurity and low birth weight are associated with an increased risk of neonatal sepsis [55]. In addition, in our study, we identified that prematurity was a risk factor for mortality in patients who already had neonatal sepsis, finding that those septic patients admitted to the NICU with a gestational age of fewer than 37 weeks had a 13.92 times greater risk of death than those who were born after. This result agrees with other studies, which attribute up to 4.6 times more risk of death to prematurity [22,56,57]. 

Furthermore, our study reports that most newborns with sepsis die within the first week of life. From the immunological point of view, it is probably because newborns with sepsis when they are premature have a higher probability of complications due to fetal adaptation and the immaturity of their system, which explains the decreased capacity of their humoral immunity and makes them unable to cope with sepsis in the first days of life [57]. This could explain why early sepsis is different from late sepsis; the immune response of the newborn in early sepsis is different from that in late sepsis. Another critical difference is the type of bacteria associated with early or late sepsis. In our study, after performing a secondary analysis of patients who presented only early sepsis, we identified that the predictors of death were similar to those of the sepsis group in general; unfortunately, we were unable to perform an analysis in the late sepsis group due to the small number of events that occurred in this group. It is suggested that future research evaluate early or late sepsis in a timely manner.

We found that those infants with neonatal sepsis who developed septic shock had a significant risk of death relative to infants who did not. Septic shock is the phase where cellular and circulatory metabolism abnormalities are complex enough to cause cardiovascular dysfunction and is associated with high hospital mortality [40,58,59]. The high mortality rate in patients who developed septic shock is probably because these patients were in advanced stages of the disease; furthermore, the newborns’ low immunity could be a determining factor in mortality. Early recognition of this complication is essential, although its identification can be tricky since it can overlap with physiological changes at birth [60]. Recent studies report that the establishment of antibiotic treatment within the first hours, especially within the first hour, for sepsis of bacterial origin turns out to be a measure that could significantly reduce mortality in these patients [48,61,62]. 

Thrombocytopenia in our study has been associated with a higher probability of death in patients with neonatal sepsis. This is similar to that reported in other studies [63,64]. This reduction in platelet levels may occur in the terminal phase of the disease and would indicate the patient’s poor prognosis. Physiopathologically, it would be associated with endothelial damage caused by inflammation and the activation of reticuloendothelial cells, which lead to the elimination of platelets. Other possible secondary pathways are low megakaryocyte count, high platelet sequestration and destruction by infection, direct cytotoxicity by bacterial endotoxins, hemophagocytic lymphohistiocytes, and disseminated intravascular coagulation (DIC) or induced by the drug treatment of neonatal sepsis (vancomycin, metronidazole, phenytoin, phenobarbital) [65,66,67].

In this study, we observed that septic neonates with elevated creatinine levels had a higher risk of death. The increase in creatinine could indicate the patient’s development of acute kidney injury (AKI) [68]. Various studies have been reported describing AKI as a probability factor for death in adult patients with sepsis and newborns [69,70,71,72]. At the renal level, the nephrons do not finish developing until week 34–36 of gestation, nephrogenesis mostly (60%) occurs at the end of pregnancy, and the number of nephrons increases by an additional 200,000 nephrons per kilogram (kg) added to the birth weight. We assume that those newborns born before 36 weeks and with low birth weight (LBW) have altered renal development in the end. In the case of neonatal sepsis, the kidney is more susceptible to damage; increasing the risk of acute renal failure increases the risk of death. This risk of death from acute renal failure is described by the authors and reinforced by our study [73,74,75]. As we observed, detecting an acute renal failure in neonatal sepsis is crucial since it leads to death. However, the current diagnostic method is recognized worldwide, and some authors describe it as suboptimal in detection. Fortunately, some studies offer a promising horizon for early detection and management, such as antioxidants and renal replacement therapy [76,77,78]. It has recently been pointed out that even the prompt administration of antibiotics cannot wholly prevent kidney tissue damage associated with pathological cascades induced by systemic inflammation and the development of oxidative stress. It is postulated that antioxidants targeting mitochondria could protect a wide range of organs from damage due to sepsis. The mitochondria-targeting antioxidant SkQR1 can be used in the septic newborn, along with conventional therapy, to treat endotoxin-induced complications such as AKI. Furthermore, it can counteract the reduced proliferative potential of renal tissue during AKI, thus improving the regenerative capacity of the organ [77].

### 4.2. Implications

The absence of rapid diagnostic methods for neonatal sepsis is detrimental. In recent years, several early biomarkers for neonatal sepsis have been detected, which offer the opportunity to start treatment quickly and on time [79]. Markers such as neutrophil gelatinase-associated lipocalin (NGAL), is a specific marker of neutrophil activity, have also been proposed as an early marker of acute kidney injury and death in patients with neonatal sepsis of bacterial origin, in addition to guiding the reduction in antibiotic therapy [80,81,82,83,84]. Interleukin 6 (IL-6), associated with C-reactive protein (CRP), plays an essential diagnostic role in neonatal sepsis in premature newborns [85,86,87] and is associated as a prognostic marker of late neonatal sepsis [88,89]. Ruetsch V. reveals that a level above 8.92 µg/L of procalcitonin (PCT) in patients with suspected sepsis is a marker of poor prognosis [89,90]. A CRP/PCT index is also described in patients with abdominal sepsis, which is directly related to death; however, we did not find studies in neonates. This index seems promising and could be considered for future studies [91].

The initiation of breastfeeding within 1 h of life in the newborn significantly affects the probability of survival [92,93,94] due to its immunological, antimicrobial, and immunomodulatory benefits in breast milk [95]. Recognizing the role that breastfeeding plays in neonatal sepsis is vital. In our study, we found that exclusive breastfeeding in neonates who developed sepsis behaved as a protective factor for death. This finding is similar to research in Ethiopia and India [96,97]. This could be explained by large amounts of immunoglobulin A-secreting antibodies produced by lymphocytes that have migrated from the mother’s intestine to the mammary glands. These antibodies are directed explicitly against the maternal intestinal microflora, the primary source of pathogens in neonatal sepsis [22,97,98].

Prematurity in the septic neonate is an important variable due to its high mortality risk. A possible treatment that is still under study is the use of probiotics. These promote the immune and epithelial response of the host, reducing inflammatory conditions in both infants and adults [99,100,101]. In recent years it has played an essential role in newborns and infants; it is reported to reduce the risk of overall death, necrotizing enterocolitis (NEC), late-onset sepsis, and food intolerance [100]. Furthermore, it is recognized that it modulates the intestine–microbiota–brain axis and has a neuroprotective function; apparently, these properties are enhanced with the joint use of exclusive breastfeeding and not with the help of infant formula [99]. On the other hand, it should also be recognized that there are side effects such as harmful metabolic activities, excessive immune stimulation, antibiotic resistance gene transfer, and gastrointestinal side effects such as intestinal gas formation. Probiotic sepsis is another possible complication, but very few cases have been described, and it can be less lethal than bacterial sepsis [102,103].

### 4.3. Limitations

This study has some limitations, which should be considered when interpreting its results. First, this was an observational and retrospective study, which reduces the possibility of controlling for various confounders inherent in the study design and the inclusion of more variables (e.g., partial parenteral nutrition and maternal characteristics). Second, the study was conducted only in a health institution, which could reduce the inference of the results. Despite this, we performed an adequate sample size calculation to answer our research question, which could have reduced this risk. Third, a significant limitation was that the study did not use the gold standard for the diagnosis of sepsis; since the number of blood cultures that managed to isolate a pathogen was minimal and this could generate an overdiagnosis of sepsis in our population to reduce this risk, we used clinical and laboratory criteria, which could improve the sensitivity in diagnosing sepsis. Finally, due to the small population that we had, we were not able to carry out analyses in subgroups (for example, patients with late sepsis) with which we would have evaluated other results; despite this, if we were able to carry out the analysis of the early sepsis subgroup, we did not find differences with the sepsis group in general.

## 5. Conclusions

To our knowledge, this is the first Peruvian study evaluating death predictors in patients with neonatal sepsis. We identified a high mortality rate, mainly developed in the first 72 h of life. The predictors of mortality in patients with neonatal sepsis identified in this study were prematurity, low birth weight, platelet levels <150,000 cells/mm^3^, creatinine >1.1 g/dL, the development of septic shock, and the use of an invasive mechanical ventilator. On the other hand, breastfeeding was associated with a lower risk of death. Early recognition of these risk factors may help identify patients with a poor prognosis early in the course of the disease and implement better strategies that could reduce neonatal sepsis-related mortality.

## Figures and Tables

**Figure 1 tropicalmed-07-00342-f001:**
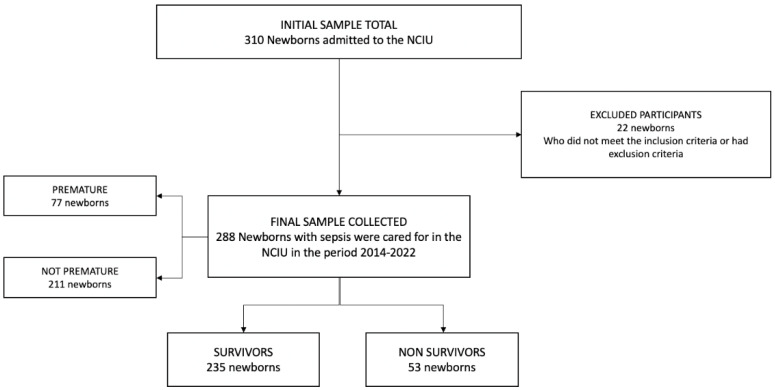
Sample selection flowchart.

**Figure 2 tropicalmed-07-00342-f002:**
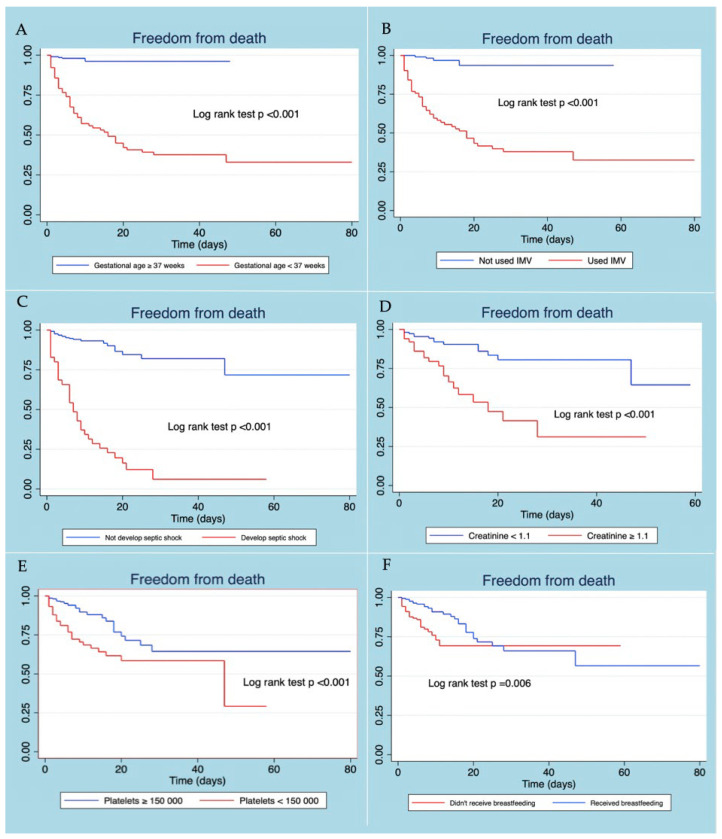
Kaplan–Meier survival curves according to gestational age (**A**), use of VMI (**B**), developed septic shock (**C**), creatinine level (**D**), platelet level (**E**), and received breastfeeding (**F**).

**Table 1 tropicalmed-07-00342-t001:** Clinical characteristics and comparison between survivors and non-survivors with neonatal sepsis.

Variables	All Patients (n = 288)	Survivors (n = 235)	Non-Survivors (n = 53)	*p*-Value
Demographic Characteristics				
Gender				0.689 ^a^
- Male	145 (50.35)	117 (80.69)	28 (19.35)	
- Woman	143 (49.65)	118 (82.52)	25 (17.48)	
Gestational age (weeks) *	39 (35–39)	39 (38–40)	30 (27–34)	<0.001 ^b^
Birth weight (grams) *	3270 (2325–3685)	3430 (2970–3820)	1160 (890–1890)	<0.001 ^b^
Type of birth				0.981 ^a^
- Vaginal	130 (45.14)	106 (80.54)	24 (18.46)	
- Cesarean section	158 (54.86)	129 (81.65)	29 (18.35)	
Reason for cesarean section				0.510 ^a^
- Fetal pathology	101 (63.92)	84 (83.17)	17 (16.83)	
- Maternal pathology	57 (36.08)	45 (78.95)	12 (21.05)	
In-hospital birth				0.554 ^a^
- No	34 (11.81)	29 (85.29)	5 (14.71)	
- Yes	254 (88.19)	206 (81.1)	48 (18.9)	
APGAR score at 1 min *	8 (6–9)	9 (7–9)	5(3–8)	<0.001 ^b^
APGAR score at 5 min *	9 (8–9)	9 (9–9)	7 (6–9)	<0.001 ^b^
APGAR score at 10 min	9 (7–9)	8 (8–9)	8 (6–8)	0.185 ^b^
Congenital disease				0.001 ^a^
- No	244 (84.72)	207 (84.84)	37 (15.16)	
- Yes	44 (15.28)	28 (63.64)	16 (36.36)	
Type of congenital disease				0.295 ^c^
- Fetal malformations	27 (61.36)	19 (70.37)	8 (29.63)	
- Genetic disease	8 (18.18)	3 (37.50)	5 (62.50)	
- Fetal infections	8 (18.18)	5 (62.50)	3 (37.50)	
- Neonatal complication	1 (2.27)	1 (100)	0 (0)	
Time to diagnosis sepsis (days) *	0 (0–1)	0 (0–1)	0 (0)	<0.001 ^b^
Hospitalized time (days) *	7 (6–14)	8 (6–14)	6 (3–12)	<0.001 ^b^
Time in NCIU (days) *	7 (5–13)	7 (5–13)	6 (3–12)	<0.004 ^b^
Classification of neonatal sepsis according to appearance				
Early sepsis				0.706 ^a^
- No	65 (22.57)	52 (80.00)	13 (20.00)	
- Yes	223 (77.43)	183 (82.06)	40 (17.94)	
Late sepsis				0.773 ^c^
- No	227 (78.82)	186 (81.94)	41 (18.06)	
- Yes	61 (21.18)	49 (80.33)	12 (19.67)	
Characteristics of NCIU management				
Use of invasive mechanical ventilation (IMV)				<0.001 ^a^
- No	206 (71.53)	201 (97.57)	5 (2.43)	
- Yes	82 (28.47)	34 (41.46)	48 (58.54)	
Central venous catheter				0.999 ^c^
- No	283 (98.26)	231 (81.63)	52 (18.37)	
- Yes	5 (1.74)	4 (80.00)	1 (20.00)	
Umbilical venous catheter				<0.001 ^a^
- No	237 (82.29)	217 (91.56)	20 (8.44)	
- Yes	51 (17.71)	18 (35.29)	33 (64.71)	
Umbilical artery catheter				<0.001 ^a^
- No	278 (96.53)	234 (84.17)	44 (15.83)	
- Yes	10 (3.47)	1 (10.00)	9 (90.00)	
Lumbar puncture				0.999 ^c^
- No	276 (95.83)	225 (81.52)	51 (18.48)	
- Yes	12 (4.17)	10 (83.33)	2 (16.67)	
Use of Breastfeeding				0.162 ^a^
- No	122 (42.36)	95 (77.87)	27 (22.13)	
- Yes	166 (57.64)	140 (84.34)	26 (15.66)	
Use of infant formula				<0.001 ^a^
- No	108 (37.5)	61 (56.48)	47 (43.52)	
- Yes	180 (62.5)	174 (96.67)	6 (3.33)	
Total parenteral nutrition				0.184 ^c^
- No	287 (99.65)	235 (81.88)	52 (18.12)	
- Yes	1 (100.00)	0 (0)	1 (100.00)	
Complications in the NCIU				
Septic shock				<0.001 ^a^
- No	253 (87.85)	231 (91.30)	22 (8.70)	
- Yes	35 (12.15)	4 (11.43)	31 (88.57)	
Breathing difficulty				<0.001 ^a^
- No	200 (69.44)	191 (95.5)	9 (4.50)	
- Yes	88 (30.56)	44 (50.00)	44 (50.00)	
Multiple organ failure				<0.001 ^a^
- No	278 (96.53)	235 (84.53)	43 (15.47)	
- Yes	10 (3.47)	0 (0.00)	10 (100.00)	
Jaundice				0.478 ^a^
- No	186 (64.58)	154 (82.80)	32 (17.20)	
- Yes	102 (35.42)	81 (79.41)	21 (20.59)	
Respiratory distress syndrome				<0.001 ^a^
- No	203 (70.49)	197 (97.04)	6 (2.96)	
- Yes	85 (29.51)	38 (44.71)	47 (55.29)	
Renal failure				0.012 ^c^
- No	282 (97.92)	233 (82.62)	49 (17.38)	
- Yes	6 (2.08)	2 (33.33)	4 (66.67)	
Seizures				0.218 ^c^
- No	279 (96.88)	229 (82.08)	50 (17.92)	
- Yes	9 (3.13)	6 (66.67)	3 (33.33)	
Vital signs				
Temperature (°C) *	36.6 (36.4–36.9)	36.7 (36.4–37)	36.2 (35.1–36.6)	<0.001 ^b^
Respiratory rate (Bpm) *	55 (50–60)	54 (48–60)	60 (52–66)	0.003 ^b^
Heart rate (bpm) *	136.5 (126–150)	135 (126–146)	146 (132–159)	0.001 ^b^
Oxygen saturation (%) *	96 (93–98)	96 (94–98)	93 (87–97)	<0.001 ^b^
Weight loss at diagnosis of sepsis (grams) *	125 (70–200)	130 (70–217.5)	90 (30–150)	0.023 ^b^
Gain weight at diagnosis of sepsis (grams) *	90 (40–290)	90 (40–295)	70 (20–100)	0.361 ^b^

* Median and interquartile range, ^a^ χ^2^ statistical test, ^b^ Mann–Whitney U statistical test, ^c^ Fisher’s exact statistical test. Bpm = breaths per minute, bpm = beats per minute.

**Table 2 tropicalmed-07-00342-t002:** Laboratory characteristics, treatment, and comparison of surviving and non-surviving patients with neonatal sepsis.

Variables	All Patients (n = 288)	Survivors (n = 235)	Non-Survivors (n = 53)	*p*-Value
Blood culture				0.796 ^a^
- Negative	262 (90.97)	213 (81.30)	49 (18.70)	
- Positive	26 (9.03)	22 (84.62)	4 (15.38)	
Isolated bacteria				0.813 ^b^
- *Klebsiella Pneumoniae* (ESBL)	16 (61.54)	12 (75.00)	4 (25.00)	
- *Staphylococcus* Coagulase-negative	5 (19.23)	5 (100.00)	0 (0.00)	
- *Escherichia coli* (ESBL)	3 (11.54)	3 (100.00)	0 (0.00)	
- *Enterobacter cloacae*	2 (7.69)	2 (100.00)	0 (0.00)	
Leukocytes (cells/mm^3^) *	17,400 (12,790–22,600)	17,600 (13,300–22,400)	14,380 (8200–25,000)	0.157 ^b^
Platelets (cells × 10^3^/L) *	200,000 (149,000–269,000)	209,000 (164,000–272,000)	145,000 (96,000–230,000)	<0.001 ^b^
Total neutrophils (cells/mm^3^) *	11,704 (7592–16,776)	12,109 (8052–16,605)	9310 (4488–17,250)	0.073 ^b^
Total band neutrophils (cells/mm^3^) *	0 (0–318)	0 (0–268)	0 (0–453)	0.253 ^b^
Ratio of immature to total neutrophils (I:T) *	0 (0–0.02)	0 (0–0.02)	0 (0–0.02)	0.185 ^b^
Hemoglobin (g/dL) *	16.6 (14.3–18.9)	16.90 (14.9–18.9)	15.3 (12.5–18.9)	0.019 ^b^
Hematocrit (%) *	49.75 (42.9–56)	50 (44–56)	47 (38.3–55.7)	0.028 ^b^
Red blood cell distribution width (%) *	15.3 (14.6–16.2)	15.3 (14.6–15.9)	16.2 (14.6–16.7)	0.550 ^b^
Albumin (g/dL) *	3.68 (2.77–4.5)	3.75 (1.39–4.16)	4.17 (3.16–5.52)	0.219 ^b^
Glucose (mg/dL) *	71.9 (57.6–98.65)	69 (57.2–88.3)	101 (63–187)	0.001 ^b^
Lactic dehydrogenase (U/L) *	1280 (462–1966)	951 (462–1966)	1468 (811.5–2705.5)	0.685 ^b^
C-reactive protein (mg/dL) *	24.70 (14.05–57)	24.89 (14.2–46.6)	22 (12.5–65.6)	0.992 ^b^
Creatinine (mg/dL) *	0.97 (0.7–1.18)	0.91 (0.69–1.09)	1.21 (0.96–1.4)	<0.001 ^b^
AST (U/L) *	63 (38–133)	71.8 (37–133)	62 (44.15–149)	0.999 ^b^
ALT (U/L) *	17.5 (8–72)	22.5 (12–153)	8.5 (6–24.5)	0.029 ^b^
Cpk-Mb (U/L) *	75.93 (25–237)	115 (32.7–262)	24 (12–25)	0.054 ^b^
Alkaline phosphatase (U/L)*	254.5 (189–342)	260.5 (213–446)	220.5 (174–260.5)	0.308 ^b^
Lactate (mmol/L) *	2.8 (2–4.8)	2.76 (2–4.5)	2.8 (2.2–6.1)	0.789 ^b^
Sodium (mmol/L) *	136 (130–142)	137.5 (131.8–143)	134.5 (123.5–138)	0.024 ^b^
Potassium (mmol/L) *	4.3 (3.76–5.01)	4.3 (3.83–4.9)	4.25 (3.35–5.65)	0.908 ^b^
Chlorine (mmol/L) *	101 (96–107)	102 (97–107)	100.5 (95–106)	0.271 ^b^
Bicarbonate (mmol/L) *	17.4 (13.5–20.1)	18.65 (15.5–20.85)	14.7 (12–17.5)	0.006 ^b^
Ph *	7.28 (7.16–7.37)	7.34 (7.25–7.41)	7.19 (7.01–7.27)	<0.001 ^b^
CO2 pressure (mmHg) *	35.2 (27–47.6)	32.8 (24.9–41.4)	46.2 (33.1–59.5)	0.002 ^b^
O2 pressure mmHg) *	64 (46–98)	61 (45–98)	64 (50.5–96.5)	0.671 ^b^
FiO2 contribution (%) *	25 (21–50)	21 (21–30)	40 (30–70)	<0.001 ^b^
PaFiO2 *	214.28 (151.42–323.80)	223.80 (171.42–432.38)	177.32 (91.44–265.71)	0.004 ^b^
Oxacillin use				0.999 ^c^
- No	264 (91.67)	215 (81.44)	49 (18.56)	
- Yes	24 (8.33)	20 (83.33)	4 (16.67)	
Use of ampicillin				0.999 ^c^
- No	17 (5.90)	14 (82.35)	3 (17.65)	
- Yes	271 (94.10)	221 (81.55)	50 (18.45)	
Use of aminoglycosides				0.778 ^c^
- No	23 (7.99)	20 (86.96)	3 (13.04)	
- Yes	265 (92.01)	215 (81.13)	50 (18.87)	
Use of cefotaxime				0.013 ^a^
- No	248 (86.11)	208 (83.87)	40 (16.13)	
- Yes	40 (13.89)	27 (67.50)	13 (32.50)	
Use of ceftazidime				0.744 ^c^
- No	273 (94.79)	223 (81.68)	50 (18.32)	
- Yes	15 (5.21)	12 (80.00)	3 (20.00)	
Use of imipenem/meropenem				<0.001 ^a^
- No	223 (77.43)	202 (90.58)	21 (9.42)	
- Yes	65 (22.57)	33 (50.77)	32 (49.23)	
Use of vancomycin				<0.001 ^a^
- No	217 (75.35)	198 (91.24)	19 (8.76)	
- Yes	71 (24.65)	37 (52.11)	34 (47.89)	
Use of metronidazole				0.040 ^c^
- No	280 (97.22)	231 (82.50)	49 (17.50)	
- Yes	8 (2.78)	4 (50.00)	4 (50.00)	
Use of clindamycin				0.316 ^c^
- No	282 (99.30)	234 (83.98)	48 (17.02)	
- Yes	2 (0.70)	1 (50.00)	1 (50.00)	
Use of fluconazole				<0.001 ^c^
- No	274 (95.14)	230 (83.94)	44 (16.06)	
- Yes	14 (4.86)	5 (35.71)	9 (64.29)	
Use of amphotericin B				0.458 ^c^
- No	285 (98.96)	233 (81.75)	52 (18.25)	
- Yes	3 (1.04)	2 (66.67)	1 (33.33)	
Use of vasopressors				<0.001 ^a^
- No	258 (89.58)	228 (88.37)	30 (11.63)	
- Yes	30 (10.42)	7 (23.33)	23 (76.67)	
Use of corticosteroids				0.063 ^c^
- No	279 (96.88)	230 (82.44)	49 (17.56)	
- Yes	9 (3.13)	5 (55.56)	4 (44.44)	
Use of diuretics				<0.001 ^a^
- No	251 (87.15)	217 (86.45)	34 (13.55)	
- Yes	37 (12.85)	18 (48.65)	19 (51.35)	
Use of bovine pulmonary surfactant				<0.001 ^a^
- No	244 (84.72)	218 (89.34)	26 (10.66)	
- Yes	44 (15.28)	17 (38.64)	27 (61.36)	
Use of phototherapy				<0.001 ^a^
- No	182 (63.19)	162 (89.01)	20 (10.99)	
- Yes	106 (36.81)	73 (68.87)	33 (31.13)	
Blood transfusion				<0.001 ^c^
- No	264 (91.67)	226 (85.61)	38 (14.39)	
- Yes	24 (8.33)	9 (37.50)	15 (62.50)	
Use of anticonvulsants				0.011 ^c^
- No	269 (93.40)	224 (83.77)	45 (16.73)	
- Yes	19 (6.60)	11 (57.89)	8 (42.11)	
Use of antipyretics				0.999 ^c^
- No	283 (98.26)	231 (81.63)	52 (18.37)	
- Yes	5 (1.74)	4 (80.00)	1 (20.00)	

* Median and interquartile range, ^a^ χ^2^ statistical test, ^b^ Mann–Whitney U statistical test, ^c^ Fisher’s exact statistical test.

**Table 3 tropicalmed-07-00342-t003:** Cox regression model 1 to find predictors of mortality in patients with neonatal sepsis.

Variable	cHR (95% CI)	*p*-Value	aHR (95% CI)	*p*-Value
Gestational age				
- Term newborn (>37)	Ref		Ref	
- Preterm newborn (<37)	19.95 (7.85–50.69)	<0.001	13.92 (1.71–113.51)	0.014
Leukocytes (cells/mm^3^)				
- Normal (<12,000)	Ref		Ref	
- Leukocytosis (>12,000)	0.39 (0.22–0.67)	0.001	0.58 (0.21–1.60)	0.298
Platelets (cells × 10^3^/L)				
- Normal (>150,000)	Ref		Ref	
- Low count (<150,000)	2.53 (1.47–4.37)	0.001	3.64 (1.22–10.88)	0.021
Ratio of immature to total neutrophils (I:T)				
- Normal (<0.12)	Ref		Ref	
- High count (≥0.12)	6.86 (2.12–22.16)	0.001	2.74 (0.28–26.46)	0.382
Glucose (mg/dL)				
- Normal (<150)	Ref		Ref	
- High count (≥150)	4.01 (2.17–7.42)	<0.001	0.60 (0.18–2.03)	0.411
Creatinine (mg/dL)				
- Normal (<1.1)	Ref		Ref	
- High count (≥1.1)	3.86 (1.94–7.68)	<0.001	3.03 (1.09–8.45)	0.034
pH				
- Normal (≥7.35)	Ref		Ref	
- Low count (<7.35)	3.24 (1.13–9.26)	0.028	2.36 (0.44–12.61)	0.315
CO2 pressure (mmHg)				
- Normal (<45)	Ref		Ref	
- High count (≥45)	2.11 (1.05–4.24)	0.035	0.99 (0.34–2.90)	0.985
PaO2/FiO2				
- Normal ≥400	Ref		Ref	
- ARDS <400	9.86 (1.34–72.60)	0.025	2.83 (0.32–25.29)	0.352

cHR: crude hazard ratio, aHR: adjusted hazard ratio, PaO2/FiO2: fraction between arterial oxygen pressure over the inspired fraction of oxygen, Ref: reference group. The proportionality of the multivariate model had a value of *p* = 0.3241.

**Table 4 tropicalmed-07-00342-t004:** Cox regression to find clinical predictors clinical of mortality in patients with neonatal sepsis.

Variable	cHR (95% CI)	*p*-Value	aHR (95% CI)	*p*-Value
Birth weight				
- Normal (≥2500 g)	Ref		Ref	
- Low birth weight (<2500 g)	15.37 (6.49–36.37)	<0.001	2.94 (1.04–8.30)	0.042
APGAR score at 1 min				
- Normal	Ref		Ref	
- Moderate	4.45 (2.33–8.47)	<0.001	0.78 (0.37–1.63)	0.505
- Low	3.47 (1.72–6.99)	<0.001	0.50 (0.18–1.42)	0.191
APGAR score at 5 min				
- Normal	Ref		Ref	
- Moderate	3.30 (1.85–5.87)	<0.001	3.21 (1.37–7.56)	0.007
- Low	1.89 (0.25–14.09)	0.533	0.39 (0.04–3.46)	0.401
Septic Shock				
- No	Ref		Ref	
- Yes	11.47 (6.63–19.85)	<0.001	4.41 (2.23–8.74)	<0.001
ARDS				
- No	Ref		Ref	
- Yes	14.40 (6.09–34.03)	<0.001	1.64 (0.52–5.21)	0.400
Use of IMV				
- No	Ref		Ref	
- Yes	19.22 (7.58–48.74)	<0.001	5.61 (1.86–16.88)	0.002
Use of CVC				
- No	Ref		Ref	
- Yes	0.41 (0.08–4.30)	0.604	1.03 (0.11–9.52)	0.979
Use of exclusive breastfeeding				
- No	Ref		Ref	
- Yes	0.47 (0.27–0.82)	0.008	0.25 (0.13–0.48)	<0.001

cHR: crude hazard ratio, aHR: adjusted hazard ratio, ARDS: acute respiratory distress syndrome, IMV: invasive mechanical ventilator, CVC: central venous catheter, Ref: reference group. The proportionality of the multivariate model had a value of *p* = 0.1882.

## Data Availability

Available upon reasonable request.

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
