# Peer review of "Predictors of Death in Patients with Neonatal Sepsis in a Peruvian Hospital"

_tropicalmed, 2022, doi:10.3390/tropicalmed7110342_

Round 1
Reviewer 1 Report
GENERAL
- Most of time, neonatal studies will provide valuable information, at least in the particular regions
- My main concerns regarding this paper are (1) The author did not analyze early and late onset sepsis separately. I suggest they add some explanations about this. (2) The calculations or statistical procedures used 2 separate tables but in the discussion the authors combine both as one
INTRODUCTION
- There were some papers regarding neonatal sepsis in Peru. I suggest the authors add some Peruvian data in the introduction.
- I also suggest the authors to add some explanation “why early and late onset sepsis was analyzed as one group”. I consider the risk factors or predictive factors between both groups were different.
METHODS
- How many neonatologists in this hospital? Were there pediatric residents also?
- Were there any inclusion and exclusion criteria, other than those already mentioned in the text?
- Clinical sepsis is similar with many other diseases of neonates. How could the author differentiate (especially if the criteria was loose)?
- What if the neonates came with other diagnosis and later in hospital sepsis appeared? Were they included also?
- If the neonates had more than 1 laboratory results on admission for the same problems (but different time), which test would be used in this study?
-
RESULTS
- There were 2 big tables in the result but in the analysis the author combined them. I suggest the author add some more explanations regarding this issue
- Table 1: “time to diagnosis sepsis” mentioned “0-1” for survivors and “0” for non survivors. However, the p value was so significant. Could the author explain?
- Microbiology: how many Acinetobacter baumanii and Group B Streptococcus in this study?
- Why in the table 3 “gestational age” was included?
- The was collinearity regarding gestational age and birth weight in some situations. They influence and depend on each other. Could the author comment on this?
DISCUSSION
- As I mentioned in the “introduction”, I suggest the author add some explanation why early and late onset sepsis were analyzed together
- Should the early onset sepsis analyzed separately, would the results be different?
REFERENCES
- Some references used capital letter only for the first word of the title. Others used capital letter for every word of the title.
- References 49 and 50: why the author used “(“ ?
- References 54: it was not clear enough
- References 72 and 74 were the same
TABLES
- Table 2: “Use of amphotericin B” the number of subject was not 235 for survivors
Author Response
Dear reviewers, here we respond to each of your observations.
Reviewer 1:
General
- Most of time, neonatal studies will provide valuable information, at least in the particular regions
- Thanks for the comment.
- My main concerns regarding this paper are (1) The author did not analyze early and late onset sepsis separately. I suggest they add some explanations about this. (2) The calculations or statistical procedures used 2 separate tables but in the discussion the authors combine both as one
- Dear reviewer, to carry out the regression and evaluate the predictors of death, we decided to carry out two regressions because we had a reduced number of events within our population (number of non-survivors = 53 patients); due to this, we included all the variables within the same model would provoke over adjustment of the variables, this was explained in the methodology section.
- On the other hand, we decided to carry out a secondary analysis to evaluate the predictors in early sepsis, and we realized that the predictors did not vary in sepsis in general. Unfortunately, we could not perform this analysis for late sepsis due to the small number of events in this population (non-survivors =12).
INTRODUCTION
- There were some papers regarding neonatal sepsis in Peru. I suggest the authors add some Peruvian data in the introduction
- You are right; we carried out the information search and added relevant data from Peru in the introduction section (Lines 48-52)
- I also suggest the authors to add some explanation “why early and late onset sepsis was analyzed as one group”. I consider the risk factors or predictive factors between both groups were different.
- Dear reviewer, we appreciate your comment; we have considered discussing the difference between early and late sepsis and their possible relationship when evaluating predictors of death in the discussion and limitations section.
METHODS
- How many neonatologists in this hospital? Were there pediatric residents also?
- In the hospital, there was one neonatologist for four years, but in the last year, two more were added. On the other hand, we have had a pediatric residency for ten years.
- Were there any inclusion and exclusion criteria, other than those already mentioned in the text?
- The inclusion and exclusion criteria we presented in the methodology were the only ones we considered to evaluate the population.
- Clinical sepsis is similar with many other diseases of neonates. How could the author differentiate (especially if the criteria was loose)?
- We took at least one acute-phase reactant to differentiate clinical sepsis as an additional criterion, as explained in the methodology section. On the other hand, we do not use the gold standard (blood culture) to diagnose neonatal sepsis because, in our hospital, we do not have an adequate culture system, and only 9% of blood cultures manage to isolate a bacterium that is often only contaminated. This is due to structural deficiencies and a lack of budget that generate the lack of trained personnel and automated laboratory equipment; this was added in the limitations section.
- What if the neonates came with other diagnosis and later in hospital sepsis appeared? Were they included also?
- If the neonate met the criteria for late-onset sepsis, it could be included in the analysis.
- If the neonates had more than 1 laboratory results on admission for the same problems (but different time), which test would be used in this study?
- We collected laboratory information when the neonate was suspected of having sepsis, and this was when the diagnosis of sepsis was confirmed. We have improved the explanation of this in the methodology section (Lines 96-97)
RESULTS
- There were 2 big tables in the result but in the analysis the author combined them. I suggest the author add some more explanations regarding this issue
- Dear reviewer, we decided to carry out two regression models due to our population's reduced number of events; this was detailed in the methodology section. On the other hand, the gestational age variable presented collinearity with the birth weight variable; due to this, we decided to divide these variables and analyze them separately. The variable birth weight and gestational age were evaluated in the first and second Cox models, respectively.
- Table 1: “time to diagnosis sepsis” mentioned “0-1” for survivors and “0” for non survivors. However, the p value was so significant. Could the author explain?
- The editor is right; we have corrected the error and recalculated the distribution and p-value.
- Microbiology: how many Acinetobacter baumanii and Group B Streptococcus in this study?
- As reported in blood cultures, no Group B streptococcus or Acinetobacter baumannii were isolated.
- Why in the table 3 “gestational age” was included?
- We performed two regression models; we separated gestational age and birth weight into different models due to the collinearity between both variables. We explained this better in the statistical analysis section.
- The was collinearity regarding gestational age and birth weight in some situations. They influence and depend on each other. Could the author comment on this?
- The reviewer is correct; we found collinearity between these two variables in the first analysis we carried out, which is why we decided to separate both variables into different models; we explained this better in the statistical analysis section.
DISCUSSION
- As I mentioned in the “introduction”, I suggest the author add some explanation why early and late onset sepsis were analyzed together
- Dear reviewer, we discuss the differences between neonates with early and late sepsis in the discussion section; on the other hand, we also add two complementary tables with the analysis of predictors of death only in patients with early sepsis.
- Should the early onset sepsis analyzed separately, would the results be different?
- We performed a secondary analysis to assess predictors of death only in patients with early sepsis; as seen in Supplementary Tables 1 and 2, we did not identify substantial variation compared to the primary analysis. On the other hand, we could not do secondary research on late sepsis because the number of events (non-survivors = 12) was too small to obtain a result.
REFERENCES
- Some references used capital letter only for the first word of the title. Others used capital letter for every word of the title.
- We have made the correction
- References 49 and 50: why the author used “(“ ?
- It was an error; we have made the correction
- References 54: it was not clear enough
- It was an error; we have made the correction
- References 72 and 74 were the same
- We have made the correction
TABLES
- Table 2: “Use of amphotericin B” the number of subject was not 235 for survivors
- It was an error; we have made the correction
We are very grateful for your comments; we believe your contribution helped improve the quality of the manuscript.

Reviewer 2 Report
“Predictors of death in patients with neonatal sepsis in a Peru-vian hospital.” is a cohort study to investigate the mortality of neonatal sepsis in Peru. Epidemiological studies of neonatal sepsis in less economically developed countries are very informative.
Major comments
1. Please describe the current status regarding neonatal sepsis in the Introduction section. Mortality, morbidity, and poor prognostic factors in developed countries are of course necessary, but please also mention mortality and morbidity in Southeast Asia, the Middle East, Africa, etc., citing the literature. Inflammatory markers are mentioned, but I don't think they are considered in this literature. Please describe the current situation with respect to the main issues that will be discussed in this study.
2. There is often no clearly defined definition of sepsis in newborns, but since this is a retrospective study, the definition of sepsis should require the fact that the bacteria were detected from the blood culture.
3. When were the blood tests and other data used? Is it at the time of admission, or at the time of diagnosis of sepsis, or the worst data? The significance of the results will depend on which data was used.
4. The results are not new findings, as the mortality rate is higher in cases with more immaturity and more advanced organ damage, which is not surprising. As the author states, the mortality rate is not much different from reports published from Africa, which seems to have a similar environment to Peru. Is the author's contention that Peru has a high mortality rate because it is not economically developed? Please state where the novelty of this study lies.
Author Response
Reviewer 2:
- “Predictors of death in patients with neonatal sepsis in a Peru-vian hospital.” is a cohort study to investigate the mortality of neonatal sepsis in Peru. Epidemiological studies of neonatal sepsis in less economically developed countries are very informative.
- You are correct in your comment.
Major comments
- Please describe the current status regarding neonatal sepsis in the Introduction section. Mortality, morbidity, and poor prognostic factors in developed countries are of course necessary, but please also mention mortality and morbidity in Southeast Asia, the Middle East, Africa, etc., citing the literature. Inflammatory markers are mentioned, but I don't think they are considered in this literature. Please describe the current situation with respect to the main issues that will be discussed in this study.
- We have modified the introduction and followed your suggestion.
- There is often no clearly defined definition of sepsis in newborns, but since this is a retrospective study, the definition of sepsis should require the fact that the bacteria were detected from the blood culture.
- You are correct in your comment; unfortunately, we could not use the gold standard (blood culture) for the diagnosis of sepsis because, in our hospital, we do not have an adequate culture system, and only 9% of the blood cultures manage to isolate a bacterium that many Sometimes it's just a contaminant. This is due to structural deficiencies and a lack of budget that generate a lack of trained personnel and automated laboratory equipment; this was added in the limitations section.
- When were the blood tests and other data used? Is it at the time of admission, or at the time of diagnosis of sepsis, or the worst data? The significance of the results will depend on which data was used.
- We collected laboratory information when the neonate was suspected of having sepsis, and this was when the diagnosis of sepsis was confirmed. We have improved the explanation of this in the methodology section (Lines 96-97)
- The results are not new findings, as the mortality rate is higher in cases with more immaturity and more advanced organ damage, which is not surprising. As the author states, the mortality rate is not much different from reports published from Africa, which seems to have a similar environment to Peru. Is the author's contention that Peru has a high mortality rate because it is not economically developed? Please state where the novelty of this study lies.
- First, this is the first informe reporting the incidence of death from neonatal sepsis in Peru; The similarity that we find with studys in Africa is highly relevant because, according to development indicators, Peru has made significant progress in different fields; but we did not find that these advances are focused on reducing neonatal death.
- On the other hand, unlike multiple reports in high-income countries in which early markers of sepsis and severity can be identified, evaluating these markers is impossible in our reality; the markers of severity in sepsis that we report here may be helpful for countries that are in conditions similar to ours since the laboratory tests evaluated are low cost.

Reviewer 3 Report
The submitted manuscript is of interest and may offer useful data for the epidemiology and the management of neonatal sepsis, in particular in Latin America.
However, I have several concerns about it.
Major concerns:
In the manuscript no data are reported about the incidence of early or late onset of the sepsis in the population study, but only comments about LOS in the manuscript. Please fix this issue.
Some studies reported that perinatal death was higher in primipares than in multipares and that the need for neonatal intensive care was significantly recorded in primiparous newborns. No data about the maternal condition has been reported and it could be of interest in defining this for the results of the present study.
lines 300-302: the authors should better describe the possible horizons of the use of antioxidants for the management of neonatal sepsis. However, they must consider that the use of probiotics is more studied and is very promising for it, and a lot of data and clinical trials are available. Please read and comment at least the following papers:
Athalye-Jape et al. doi: 10.1111/1751-7915.13357
Kulkarni et al. doi: 10.1007/s00431-022-04452-5
Santacroce et al. doi: 10.1080/14787210.2019.1645597
Schüller et al. doi: 10.3389/fped.2018.00199
Sinha et al. doi: 10.1186/s13063-021-05193-w
van den Akker et al. doi: 10.1097/MPG.0000000000002655
Zhang et al. doi: 10.1097/MD.0000000000002581
Minor concerns:
line 42: what is "local microbiology"?
line 106: "Additional reference test" should be a sub-paragraph
line 119 "Complications during NCIU stay" should be a sub-paragraph
line 123 "Treatment Received" should be a sub-paragraph
line 133: change chi2 as "χ²" or "Pearson's chi-squared test"
lines 154-155: the text seems to be in contrast with lines 79-80
line 225: gestational age <37 years?? Is it referred to the maternal or to the fetal age?
At last, an English language revision and typos' corrections are required.
Author Response
Dear reviewer, here we respond to each of your observations.
Reviewer 3
- The submitted manuscript is of interest and may offer useful data for the epidemiology and the management of neonatal sepsis, in particular in Latin America.
- Thank you very much dear reviewer.
Major concerns:
- In the manuscript no data are reported about the incidence of early or late onset of the sepsis in the population study, but only comments about LOS in the manuscript. Please fix this issue.
- In Table 1, we report the number of patients who developed early and late sepsis cases.
- Some studies reported that perinatal death was higher in primipares than in multipares and that the need for neonatal intensive care was significantly recorded in primiparous newborns. No data about the maternal condition has been reported and it could be of interest in defining this for the results of the present study.
- Dear reviewer, this variable was not considered when carrying out the protocol; it is a limitation of the study that will be added.
- lines 300-302: the authors should better describe the possible horizons of the use of antioxidants for the management of neonatal sepsis. However, they must consider that the use of probiotics is more studied and is very promising for it, and a lot of data and clinical trials are available. Please read and comment at least the following papers:
- You are correct in your comment; we made a more extensive discussion on these points, as seen in the lines (341-351) of the discussion.
Athalye-Jape et al. doi: 10.1111/1751-7915.13357 Kulkarni et al. doi: 10.1007/s00431-022-04452-5 Santacroce et al. doi: 10.1080/14787210.2019.1645597 Schüller et al. doi: 10.3389/fped.2018.00199 Sinha et al. doi: 10.1186/s13063-021-05193-w van den Akker et al. doi: 10.1097/MPG.0000000000002655 Zhang et al. doi: 10.1097/MD.0000000000002581
Minor concerns
- line 42: what is "local microbiology"?
- They are the most common bacteria that occur in a specific area.
- line 106: "Additional reference test" should be a sub-paragraph
- We have corrected it.
- line 119 "Complications during NCIU stay" should be a sub-paragraph
- We have corrected it.
- line 123 "Treatment Received" should be a sub-paragraph
- We have corrected it.
- line 133: change chi2 as "χ²" or "Pearson's chi-squared test"
- We have corrected it.
- lines 154-155: the text seems to be in contrast with lines 79-80
- We have corrected it.
- line 225: gestational age <37 years?? Is it referred to the maternal or to the fetal age?
- It refers to fetal age; we corrected the error.
- At last, an English language revision and typos' corrections are required.
- Dear reviewer, we performed a redaction review throughout the entire manuscript.
We are very grateful for your comments; we believe your contribution helped improve the quality of the manuscript.

Round 2
Reviewer 2 Report
I think it is necessary to revise the English after receiving the English check, but I don't think it is necessary to revise the manuscript any further.
Reviewer 3 Report
The authors have improved the quality of the manuscript according to the previous comments.